# Association of Serum Levels and Immunohistochemical Labelling of Des-Gamma-Carboxy-Prothrombin in Patients Undergoing Liver Transplantation for Hepatocellular Carcinoma

**DOI:** 10.3390/diagnostics14090894

**Published:** 2024-04-25

**Authors:** Suzanne Chabert, Samuele Iesari, Geraldine Dahlqvist, Mina Komuta, Pamela Baldin, Evaldo Favi, Laurent Coubeau

**Affiliations:** 1Department of Hepatogastroenterology, Cliniques Universitaires Saint-Luc, 1200 Brussels, Belgium; suzanne.chabert@gmail.com (S.C.); geraldine.dahlqvist@saintluc.uclouvain.be (G.D.); 2General Surgery and Kidney Transplantation, Fondazione IRCCS Ca’ Granda Ospedale Maggiore Policlinico, 20122 Milan, Italy; samuele.iesari@policlinico.mi.it (S.I.); evaldo.favi@unimi.it (E.F.); 3Department of Pathology, Keio University, Tokyo 223-8522, Japan; mina.komuta.pathology@gmail.com; 4Department of Pathology, Université Catholique de Louvain, 1200 Brussels, Belgium; pamela.baldin@saintluc.uclouvain.be; 5Pôle de Chirurgie Expérimentale et Transplantation, Institut de Recherche Expérimentale et Clinique, Université Catholique de Louvain, 1200 Brussels, Belgium; 6Service de Chirurgie et Transplantation Abdominale, Cliniques Universitaires Saint-Luc, 1200 Brussels, Belgium

**Keywords:** hepatocellular carcinoma, liver transplantation, des-gamma-carboxy-prothrombin

## Abstract

Hepatocellular cancer (HCC) is one of the main reasons for liver transplantation (LT). Biomarkers, such as alpha-foetoprotein (AFP) and Des-gamma-carboxy-prothrombin (DCP), can be helpful in defining the recurrence risk post LT. This study aims to evaluate the association between the intensity of DCP immunohistochemical labelling and serum DCP levels in patients undergoing LT for HCC. We carried out a prospective monocentric study including patients who all underwent LT for cirrhosis between 2016 and 2018 and all fell under the Milan criteria. The accepted diagnostic criteria for HCC were contrast-enhanced imaging and histology. Thirty-nine patients were followed for a median of 21 months, with HCC lesions categorized into negative, focally positive, and diffusely positive groups based on DCP immunohistochemistry. The serum DCP levels were significantly higher in the positive groups (258 mAU/mL for the focally and 257 mAU/mL for the diffusely positive) than in the negative group (48 mAU/mL) (*p* = 0.005) at diagnosis and at the time of liver transplantation (220 mAU/mL for the diffuse positive group). Microvascular invasion (58.8% vs. 19.0% for the diffusely positive and negative groups, respectively, *p* < 0.001) and lesion size (20 mm in the diffusely labelled group versus 12 mm in the other groups, *p* = 0.002) were significantly correlated with DCP labelling. Late recurrence occurred only in the positive groups; in the negative group, it occurred within the first 3 months after transplantation. DCP labelling in liver lesions correlates with serum levels and a more aggressive tumour profile. Further investigation is needed to determine if highly DCP-labelled tumours allow for the better selection of high-risk patients before LT.

## 1. Introduction

HCC is the fifth most common cancer worldwide and the third in terms of mortality [1,2,3,4,5]. The incidence of this primary liver tumour is increasing while its overall prognosis remains poor unless an early diagnosis is made [1,2,3,4,5,6,7]. The value of specific HCC biomarkers, such as AFP, is currently well established for the purposes of screening, prognosis and waiting list registration for liver transplantation (LT) [1,2,3,4,5,6,7,8,9,10,11,12,13,14,15,16,17]. Similarly, serum DCP, also known as protein induced by vitamin K absence or antagonist II (PIVKA-II), has been validated as a biomarker for the diagnosis of HCC over recent years [1,2,3,4,5,6,7,8,9,10,11,12,14,15,16,17,18,19,20,21,22]. 

DCP is a protein induced by the lack of vitamin K (VK) or the administration of VK antagonists. The deficiency in VK hinders the carboxylation of 10 glutamic acid residues in the N-terminal portion of the protein, resulting in an abnormal non-functional prothrombin. Well-functioning hepatocytes carry out post-translational carboxylation before releasing prothrombin into the peripheral blood. The carboxylation converts specific amino-terminal glutamic acid residues to gamma-carboxyglutamic acid. Most HCC cells do not express the specific VK-dependent carboxylase. Therefore, this non-carboxylated form of prothrombin has taken on the role of a serum biomarker for HCC [5,8,9,14,17,19,22,23].

In recent years, DCP has been validated for HCC diagnosis, especially in Asia. Recent studies have suggested that the diagnostic accuracy of DCP is higher (sensitivity, 86–89%; specificity, 87–93%) compared to that of AFP [1,5,6,7,8,17]. Therefore, DCP is a candidate biomarker for use in the process of HCC screening. There is evidence that higher serum levels of AFP and DCP at the time of diagnosis are correlated with worse oncological outcomes [1,2,3,4,5,7,8,9,10,12,13,14,15,16,17,18,19,20,21,23,24,25,26,27].

The treatment of HCC remains a challenge, with overall unsatisfactory survival rates of 20 and 5% for the 1- and 3-year survival, respectively [6]. Nonetheless, LT is the best curative treatment for patients with liver-confined HCC. It is thus necessary to find biomarkers that can best predict the risk of recurrence and survival, to help select appropriate candidates for LT. Several studies have shown a correlation between serum DCP levels and the presence of microvascular invasion (MVI), and of intrahepatic metastasis, the number of tumours and poor tumour differentiation [1,2,3,4,5,7,8,9,10,12,13,14,15,16,17,18,19,20,21,23,24,25,26,27]. These factors are associated with a higher risk of recurrence of HCC [2,3,5,7,8,9,10,11,12,13,14,15,16,18,19,20,21,23,24,25,26,27]. On the contrary, it is less clear whether and to what extent tissue DCP labelling reflects circulating DCP levels, or whether it is associated with worse HCC histological features and, in the end, HCC aggressiveness.

Therefore, the primary objective of our study was to evaluate the association of liver labelling with serum levels of DCP and worrisome histological features of HCC. Then, we aimed to investigate whether positive labelling was linked to HCC recurrence after LT. With this perspective, we also want to evaluate if there is a place for the use of DCP as a screening tool for the management of patients with HCC, a matter of relevance to public health decision-making.

## 2. Method

### 2.1. Patients 

Thirty-nine patients that were undergoing LT at Saint-Luc University Hospital were prospectively included between 2016 and 2018. 

Inclusion criteria were age > 18 years and LT for cirrhosis complicated by radiological and histological diagnosis of HCC within Milan criteria. Patients receiving warfarin or vitamin K were excluded because serum DCP levels are highly influenced by these medications, irrespective of the presence or absence of HCC.

### 2.2. Evaluation of HCC

HCC was diagnosed based on clinical (serum DCP and AFP) and radiological data. HCC diagnosis was confirmed by histological examination of the operative specimen.

### 2.3. Blood Samples

Serum DCP and AFP levels were checked at the time of diagnosis and at LT. All patients were followed up for two years and monitored by using serum AFP and DCP after LT (at 6 months, one and two years). DCP was measured with AIA-Pack^®^ PIVKA-II (Tosoh Europe BV, Amsterdam, The Netherlands) on the AIA-CL analyzer (Tosoh Europe BV, Amsterdam, The Netherlands) by means of a chemiluminescent enzyme immunoassay (CLIA), based on the two-step sandwich method. We used the MU-3 mouse anti-DCP monoclonal antibody. AFP was determined on Cobas e602 (Elecsys AFP, Roche, Basel, Switzerland), a module that performs sandwich immunoassays based on electrochemiluminescence (ECLIA). Serum levels of DCP and AFP were compared with the clinical features such as intrahepatic recurrence and distant metastasis of HCC, and the pathological aspects of the liver specimen such as tumour size, histologic grade, capsule, MVI, intrahepatic metastasis, and DCP and AFP immunoreactivity.

### 2.4. Tissue Samples

Histological diagnosis of HCC was made on the operative explant liver according to the World Health Organization (WHO) criteria [28]. 

We examined thirty-nine livers and we obtained five 5 µm thick slides of every lesion-containing paraffin block.

### 2.5. Immunohistochemistry 

Immunohistochemistry was performed on representative slides from each case, by using paraffin-embedded sections of 50 HCCs, with antibodies against DCP and AFP (A0008 Dako, Agilent, Santa Clara, CA, USA). Automated staining and detection were performed using the BenchMark XT system along with the ultraView Universal DAB Detection Kit (Ventana, Roche Diagnostics International AG, Rotkreuz, Switzerland). Two independent pathologists carried out a double-blind assessment of the degree of labelling. Immunolabelling was considered positive if more than 5% of tumour cells were stained according to the proper pattern of reactivity.

### 2.6. Statistical Analysis

Continuous data were reported as medians and interquartile ranges and tested with the Kruskal–Wallis test, where appropriate. Post hoc pairwise comparisons were carried out by means of the Dunn’s multiple comparisons test and reported if statistically significant. Binomial variables were reported as percentages and tested with the Χ^2^ test, where appropriate. The time to recurrence was analysed with the Kaplan–Meier method and compared with the log-rank test. The significance of statistical tests was taken at a *p*-value < 0.05. Analyses were run using SPSS (version 25.0; IBM Corp., Armonk, NY, USA) and Prism (version 9.5, GraphPad Software, Boston, MA, USA).

## 3. Results 

### 3.1. Overview

The baseline characteristics of the population are summarised in Table 1 and Appendix A. 

The HCC lesions were stratified into three groups depending on the DCP labelling: negative labelling, focally positive labelling, and diffusely positive labelling (Figure 1).

Eighty-two per cent of the patients were male with a median age of 67 (IQR 63–71).

Alcohol-related liver disease was the most frequent aetiology of the underlying disease (59%) followed by HCV (25.6%). Respectively, 83% of the patients with HBV-related cirrhosis and 80% of the patients with HCV-related cirrhosis showed a sustained viral response. 

As for the general comorbidities, most of our population were overweight, as shown by a median BMI of 27, while 38% of the patients had diabetes and 2% had chronic pancreatitis. 

We evaluated the severity of the cirrhosis by means of the Child–Pugh score (CPS) and the MELD score. The median CPS was 6 (IQR 5–7) and the median MELD amounted to 10 (IQR 8–14), with no significant differences between the three groups (*p* = 0.230 and *p* = 0.195, respectively).

### 3.2. Oncological Characteristics

The oncological characteristics are also included in Table 1. 

Eighty-seven per cent of the patients had undergone a locoregional treatment for HCC before LT, the majority having received at least two treatments. Trans-arterial chemoembolization (TACE) was the most frequent treatment (72%) followed by radiofrequency ablation (RFA) (18%). The median time between the LT and the last treatment was 5 months (IQR 2–9), with a longer period without treatment in the negative labelling group (8 months). 

### 3.3. AFP and DCP at Diagnosis and at Transplantation

The serum DCP levels at diagnosis were significantly higher in the patients harbouring diffusely positive lesions (257.3 mAU/mL) compared to the candidates with negatively labelled lesions (47.5 mAU/mL, *p* = 0.004, Table 1). At the time of LT, the serum DCP was significantly higher in the diffusely positive group (220.2 mAU/mL) compared to the candidates with negatively labelled lesions (32.0 mAU/mL, *p* < 0.001, Table 1). 

With regard to the discriminative power of the serum markers for recurrence, this study was underpowered to establish whether the serum DCP, either at cancer diagnosis or at LT, has a discriminative power for postoperative cancer recurrence (Figure 2). 

### 3.4. Transplantation Characteristics

We reported the type of donor and the cold ischemia time (CIT); 67% were donors after brain death (DBD) and the median CIT was 463 min (IQR 398–544, Appendix A).

### 3.5. Tumour Characteristics and Biomarker Levels

In Table 2, we report the tumour characteristics per intensity of DCP labelling per each single liver lesion of HCC (119 nodules, overall) found in the surgical specimens of the 39 LT recipients.

When looking at each individual lesion, we observed a significant difference in the presence of MVI in the diffusely positive lesions compared to the negative labelling group (58.8% vs. 19% *p* < 0.001). The size of the lesion significantly correlated with the DCP labelling. The median maximum diameter was 20 mm (IQR 12–26) in the diffusely positive labelling group compared to 12 mm (IQR 7–18) in the negative labelling group (*p* = 0.003) and to 12 mm (IQR 8–18) in the focally positive labelling group (*p* = 0.023, Table 2). 

### 3.6. Risk of Recurrence

Of the thirty-nine patients, six experienced a recurrence after the LT (15%).

When looking at the recurrence rate, the two cases of recurrence in the negative labelling group occurred within the first 6 months after LT.

Late recurrence only occurred in the two other groups (Figure 3). 

## 4. Discussion 

Prothrombin is a vitamin-K-dependent coagulation factor and is synthesised by the liver. The gamma-glutamyl carboxylase is an enzyme that requires vitamin K as a co-factor to transform glutamic acid residues (Glu) into the gamma-carboxylated residues (Gla) of standard prothrombin. Glu can be found on the prothrombin precursor. The loss of this gamma-carboxylation reaction, for example, because of vitamin K deficiency, leads to the upholding of some or all of the Glu residues, leading to the release of an abnormal prothrombin, i.e., DCP. The latter form loses its capacity to interact with other coagulation factors [5,8,9,14,17,19,22,23]. 

The molecular mechanisms behind the production of DCP by malignant HCC cells are not fully understood. Several studies have been performed on this matter and three hypotheses have been highlighted, as described in an article by Inagaki et al. [9]. The first possible explanation is that the activity of the gamma-glutamyl carboxylase decreases in cancer tissue because of the expression of an abnormal, less functional gamma-glutamyl carboxylase protein. A second mechanism might involve altered vitamin K metabolism, which is responsible for vitamin overconsumption and decreased availability. Some studies have shown that, in the presence of vitamin K, HCC cells do not produce DCP. Consequently, the administration of vitamin K reduces the serum levels of DCP. An in vitro study conducted by Murata et al. [19], demonstrated that changes in cytoskeletal filaments are observed during the epithelial-to-mesenchymal transition. These changes actually hamper the endocytosis of vitamin K, therefore impacting DCP production. A third hypothesis suggests that DCP is the result of the overexpression of the prothrombin precursor in cancer tissue. In summary, a number of factors interfere with the production of DCP at a cellular level in HCC tumoral cells. 

Based on in vitro evidence, the serum DCP levels are associated with the presence of HCC. With regard to this condition, early diagnosis is the key for a better prognosis but remains a challenge. While our study was underpowered to confirm whether serum DCP levels discriminate for postoperative cancer recurrence, a number of experiences, mostly from Asia, have clearly confirmed the usefulness of serum DCP as an HCC diagnostic as well as prognostic marker [3,16,24]. In addition, serum DCP levels correlate with tumoral aggressiveness features, such as cellular dedifferentiation, the number of lesions, intra-hepatic metastasis, and, principally, microvascular invasion (MVI) [1,2,3,4,5,7,8,9,10,12,13,14,15,16,17,18,19,20,21,23,24,25,26,27]. From the European experience, a few studies have explored the usefulness of serum DCP as a prognostic tool [21]. Overall, however, there is scarce information about the relationship between tissue immunohistochemical labelling of DCP and HCC recurrence. With our experience, we tried, then, to shed some light in this field.

In our study, we stratified the participants into three groups. This stratification is built on the immunohistochemical DCP labelling profile of liver specimens obtained during LT. We found out that there is a significant correlation between the serum levels of DCP and diffuse tissue immunohistochemical labelling. Indeed, at the time of first HCC diagnosis, the diffusely labelled group showed the highest serum DCP levels compared to the negative group. At the time of transplantation, DCP levels were significantly higher in the diffusely positive group compared to the negative group. Regarding the tumour characteristics, we noticed a significant difference in the presence of MVI in the two positive groups compared to the negative one. In essence, our study confirmed that the circulating DCP ostensibly is related to tissue DCP, and, notably, that the latter is linked to the presence of MVI. Thus, serum and immunohistochemical DCPs are good candidates as prognostic factors for the presence of MVI, which is largely established as an independent risk factor for recurrence. 

In the study by Inagaki et al., they propose two of the possible mechanisms linking DCP and tumoral progression [9]. The first option assumes that DCP affects the proliferation of HCC cells by structurally mimicking the hepatocyte growth factor. In this way, DCP activates the Met-JAK-STAT pathway that fosters tumoral cellular proliferation. The second possible explanation states that DCP increases angiogenesis. A study performed by using human umbilical vein endothelial cells has shown that DCP leads to the overexpression of EGFR and VEGF [22]. The extra production of DCP by tumoral cells would therefore increase the vascularisation of the surrounding tissue. By means of a combination of the two proposed mechanisms, DCP is likely to confer cancer cells survival and a proliferation advantage, playing a key role in oncogenesis and cancer aggressiveness. 

Regarding the recurrence rate, we had six cases of recurrence. Very early recurrence, i.e., within the first six months after LT, happened in two patients who harboured negatively labelled lesions. While, at this stage, we would not assume that only tissue DCP-labelled cancers would recur, it might be of interest to know that these two patients had serum AFP levels at LT of 211 and 381 ng/mL, extremely high compared to the rest of the patients, who did not show tissue DCP labelling and recurrence (median AFP level of 4.7 ng/mL (IQR 2.7–5.6)). Moreover, microvascular invasion was detected in the specimens of these two patients, once more confirming that we dealt with two advanced cases. While this might reduce the value of tissue DCP labelling for the prediction of very early recurrence, we could not establish whether these two cases had been, in fact, incorrectly selected for LT, in terms of deep pre-transplant locoregional cancer treatment and evaluation of cancer spread status.

Conversely, later on, recurrence only occurred in the two groups that showed positively labelled lesions. The use of biological markers, only some of which are serum and tissue DCP, to predict post-LT HCC recurrence raises the debated question of patient selection for LT. Traditionally, morphological criteria obtained on cross-sectional imaging, of which the Milan criteria are only an example, have been dominantly used in national protocols for the registration of candidates on waiting lists. It is known that the Milan criteria are based on the number and size of lesions, which casts out from a curative pathway a proportion of patients who are unlikely to experience recurrence and includes patients with a bad prognosis and misleading imaging. The transplantation community has been studying for more than a decade how to refine selection by introducing biological and dynamic criteria into the algorithm. Serum DCP has been already confirmed as a predictive tool, identifying patients outside the Milan criteria but with good disease-free survival after LT. Based on this evidence, three Japanese centres have set up new extended criteria: the “Tokyo criteria” (≤5 tumours, each with a diameter ≤5 cm, in combination with serum levels of AFP and of DCP, respectively, of ≤250 ng/mL and ≤450 mAU/mL), the “Kyoto criteria” (≤10 tumours, each with a diameter ≤ 5 cm, in combination with serum DCP ≤ 400 mAU/mL), and, lastly, the “Kyushu criteria” (≤5 cm tumours, in combination with serum DCP ≤ 300 mAU/mL) [9,10,11,12,13,16,18,24,25,26]. Among others, Fujiki et al. have demonstrated the good performance of these new criteria [18]. In their experience, patients outside the Milan criteria but within the Kyoto criteria were comparable to patients within the Milan criteria, by showing a five-year recurrence rate of 4% vs. 7% and a 5-year survival of 89% vs. 78%, respectively.

In our study, we approached the question about the function of tissue DCP as a meaningful selection marker and of the role of serum and tissue DCP, combined, in this context. We obtained some evidence in this area that merit further investigation. One of the limitations of our experience, besides the reduced sample size, is the fact that we could implement immunohistochemical labelling only on surgical specimens, while the main interest of this marker would be at the beginning of the selection process and not at the end. Nonetheless, preoperative needle biopsy is usually regarded as being risky and poorly informative. HCC is indeed a very heterogeneous disease (intra and between lesions), and needle biopsy poses the risk of tumour dissemination. However, DCP immunohistochemistry might have a role in the association of functional imaging techniques, such as PET-CT, serum DCP measurements, and tissue labelling. In this way, needle biopsy might be reserved for high-risk patients. This strategy would help include patients for LT instead of simply excluding them on the grounds of imaging and serum markers. Tissue DCP might integrate a composite score embracing serum levels and immunohistochemistry for DCP. All of these postulates have to be explored in further studies. Our results confirm the place of DCP in public health as a screening tool for the management of patients with HCC, in association with imaging and other biological markers. 

In addition, with our work, we raise the question of the role and relevance of tissue DCP labelling in primary liver diseases and in primary liver cancers other than hepatocellular carcinoma, which also should be the focus of further investigations.

In conclusion, serum DCP is established as a diagnostic marker of HCC, alone or in association with other biological markers, namely, AFP, in patients at risk. We found a significant correlation between serum DCP levels, the immunohistochemical DCP labelling and poor tumoral prognostic factors, especially MVI. This suggests a direct role of tissue DCP as a prognostic marker. Further research is warranted to assess whether tissue DCP labelling can become a selection tool to give access to LT to some patients that would be otherwise excluded on the grounds of imaging criteria. The first step includes a larger-scale study to validate our results and a second step involves an evaluation of DCP labelling at more time points to capture the significance of the dynamic evolution of this tissue marker.

## Figures and Tables

**Figure 1 diagnostics-14-00894-f001:**
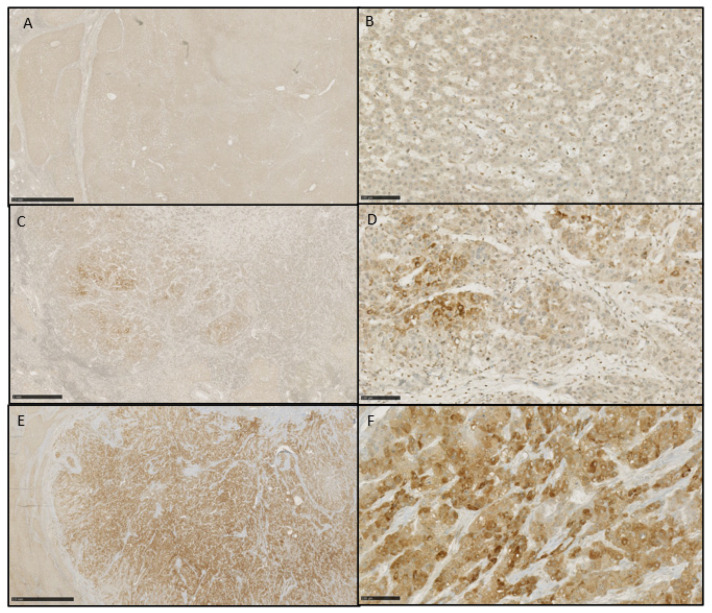
Immunochemistry labelling of DCP on liver sections after LT. In (**A**,**B**), the staining is completely negative ((**A**) DCP 1.25×, (**B**) DCP 20×). In (**C**,**D**), the staining is focal ((**C**) DCP 2.5×, (**D**) 20×). In (**E**,**F**), the staining is diffusely positive ((**E**) 1.25×, (**F**) 20×).

**Figure 2 diagnostics-14-00894-f002:**
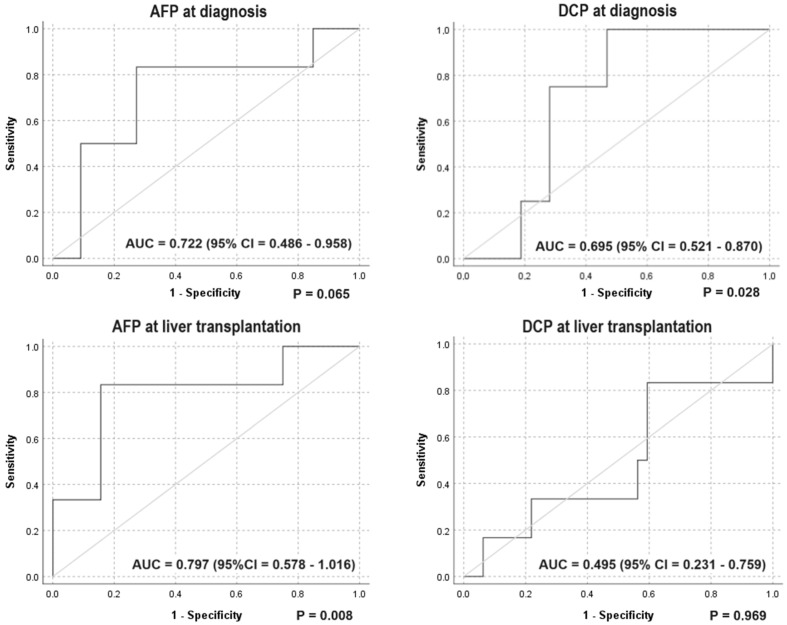
ROC curves for the discriminative power of serum HCC markers for postoperative recurrence.

**Figure 3 diagnostics-14-00894-f003:**
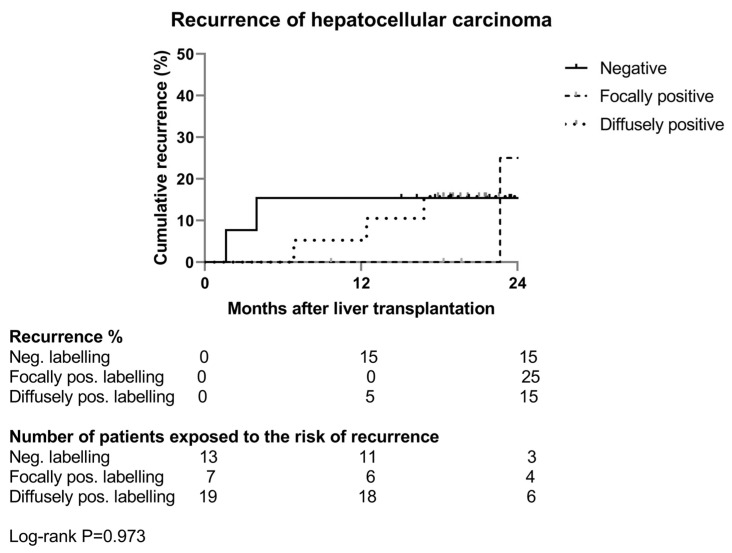
Kaplan–Meier analysis of hepatocellular carcinoma recurrence per intensity of des-γ-carboxy prothrombin labelling of hepatocellular carcinoma. Uninterrupted line: patients with negative labelling; dashed line: patients with focally positive labelling; dotted line: patients with diffusely positive labelling.

**Table 1 diagnostics-14-00894-t001:** Demographics of liver transplant recipients per intensity of des-γ-carboxy prothrombin labelling of hepatocellular carcinoma ^a^.

	Whole Population (*n* = 39)	Negative Labelling (*n* = 13)	Focally Positive Labelling (*n* = 7)	Diffusely Positive Labelling (*n* = 19)	*p* ^b^
	Medians (IQR) or *n* (%)				
Age (years)	67 (63–71)	67 (62–72)	67 (54–71)	68 (65–72)	0.725
BMI (kg/m^2^)	27.9 (24.8–31.2)	29.7 (25.2–31.4)	30.2 (26.9–31.2)	27.6 (24.0–30.0)	0.387
Gender (male)	32 (82.1)	9 (69.2)	6 (85.7)	17 (89.5)	0.329
Underlying liver disease					
HBV	6 (15.4)	4 (30.8)	0 (0.0)	2 (10.5)	0.137
Sustained viral response	5/6 (83.3)	4/4 (75.0)	0/0 (0.0)	2/2 (100.0)	0.439
HCV	10 (25.6)	4 (30.8)	3 (42.9)	3 (15.8)	0.327
Sustained viral response	8/10 (80.0)	4/4 (100.0)	2/3 (66.7)	2/3 (66.7)	0.435
Alcohol-related disease	23 (59.0)	6 (46.2)	5 (71.4)	12 (63.2)	0.480
Haemochromatosis	2 (5.1)	0 (0.0)	0 (0.0)	2 (10.5)	0.330
NASH	5 (12.8)	1 (7.7)	1 (14.3)	3 (15.8)	0.791
SBC	1 (2.6)	0 (0.0)	0 (0.0)	1 (5.3)	0.583
Cryptogenic	1 (2.6)	0 (0.0)	0 (0.0)	1 (5.3)	0.583
Comorbidities					
Diabetes mellitus	15 (38.5)	6 (46.2)	3 (42.9)	6 (31.6)	0.683
Chronic pancreatitis	2 (5.1)	0 (0.0)	0 (0.0)	2 (10.5)	0.330
Liver disease severity					
CTP score	6 (5–7)	5 (5–7)	5 (5–8)	6 (5–8)	0.230
MELD score	10 (8–14)	9 (8–12)	9 (7–17)	11 (10–15)	0.195
Oncologic variables					
Satellitosis	1 (2.6)	0 (0.0)	0 (0.0)	1 (5.3)	0.583
Previous LRT	34 (87.2)	13 (100.0)	6 (85.7)	15 (78.9)	0.215
Number of LRTs	2 (1–4)	3 (2–4)	2 (1–4)	2 (1–3)	0.291
Resection	4 (10.3)	2 (15.4)	0 (0.0)	2 (10.5)	0.556
TACE	28 (71.8)	11 (84.6)	5 (71.4)	12 (63.2)	0.416
RFA	7 (17.9)	3 (23.1)	0 (0.0)	4 (21.1)	0.389
PEI	4 (10.3)	1 (7.7)	1 (14.3)	2 (10.5)	0.897
TAE	2 (5.1)	1 (7.7)	0 (0.0)	1 (5.3)	0.758
External RT	3 (7.7)	3 (23.1)	0 (0.0)	0 (0.0)	**0.039**
SIRT	2 (5.1)	0 (0.0)	0 (0.0)	2 (10.5)	0.330
Last-LRT-to-LT interval (months)	5 (2–9)	8 (3–11)	2 (1–5)	4 (1–7)	0.084
Serum AFP at diagnosis (ng/mL)	8.4 (4.2–40.2)	4.2 (3.2–72.5)	50 (5.8–288.8)	8.7 (6.1–14.3)	0.235
Serum DCP at diagnosis (mAU/mL)	169.0 (48.4–296.0)	47.5 (29.6–189.4)	258.0 (102.2–298.4)	257.3 (94.2–649.1)	**0.005 ^c^**
Serum AFP at LT (ng/mL)	6.8 (4.3–15.0)	5.3 (3.1–7.7)	8.8 (4.1–13.8)	8.2 (4.9–16.9)	0.472
Serum DCP at LT (mAU/mL)	57.5 (31.3–225.8)	32.0 (27.5–52.7)	55.3 (29.7–119.6)	220.2 (91.2–451.7)	**<0.001 ^d^**
Follow-up (months)	21 (19–25)	22 (18–23)	24 (18–25)	21 (19–25)	0.921

The values of *p* that are statistically significant are reported in bold. ^a^ Patients are classified according to the most intense labelling; ^b^ Kruskal–Wallis tests for quantitative variables and Χ^2^ tests for nominal variables; ^c^ Dunn’s multiple comparisons test for negative labelling vs. diffusely positive labelling *p* = 0.004; ^d^ Dunn’s multiple comparisons test for negative labelling vs. diffusely positive labelling *p* < 0.001. Abbreviations: AFP, α-foetoprotein; CTP, Child–Turcotte–Pugh score; DCP, des-γ-carboxy prothrombin; IQR, interquartile range; LRT, locoregional treatment; LT, liver transplantation; MELD, model for end-stage liver disease; NASH, non-alcoholic steatohepatitis; PEI, percutaneous ethanol injection; RT, radiation therapy; SBC, secondary biliary cirrhosis; SIRT, selective internal radiation therapy; TACE, trans-arterial chemoembolization; TAE, trans-arterial embolization.

**Table 2 diagnostics-14-00894-t002:** Distribution of microvascular invasion and tumour characteristics per intensity of des-γ-carboxy prothrombin labelling in liver lesions of hepatocellular carcinoma.

	All Lesions (*n* = 119)	Negative Labelling (*n* = 58)	Focally Positive Labelling (*n* = 27)	Diffusely Positive Labelling (*n* = 34)	*p* ^a^
	Medians (IQR) or *n* (%)				
Maximum diameter (mm)	12 (9–21)	12 (7–18)	12 (8–18)	20 (12–26)	**0.002 ^b^**
Microvascular invasion (presence)	44 (37.0)	11 (19.0)	13 (48.1)	20 (58.8)	**<0.001**
Differentiation					
No tumour residue	20 (16.8)	20 (34.5)	0 (0.0)	0 (0.0)	**<0.001**
Low-grade dysplasia	2 (1.7)	2 (3.4)	0 (0.0)	0 (0.0)	0.343
High-grade dysplasia	1 (0.8)	1 (1.7)	0 (0.0)	0 (0.0)	0.588
Well-differentiated tumour	17 (14.3)	12 (20.7)	3 (11.1)	2 (5.9)	0.127
Moderately differentiated tumour	65 (54.6)	17 (29.3)	21 (77.8)	27 (79.4)	**<0.001**
Poorly differentiated tumour	10 (8.4)	5 (8.6)	3 (11.1)	2 (5.9)	0.763
Features of CCC or mixed HCC-CCC	4 (3.4)	1 (1.7)	0 (0.0)	3 (8.8)	0.103

The values of *p* that are statistically significant are reported in bold. ^a^ Kruskal–Wallis tests for quantitative variables and Χ^2^ tests for nominal variables; ^b^ Dunn’s multiple comparisons test for negative labelling vs. diffusely positive labelling *p* = 0.003 and for focally positive labelling vs. diffusely positive labelling *p* = 0.023. Abbreviations: CCC, cholangiocellular carcinoma; HCC, hepatocellular carcinoma.

## Data Availability

The data presented in this study are available upon reasonable request from the corresponding author. The data are not publicly available because the participant did not give written consent for their data to be shared publicly.

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
