# Peer review of "Association of Serum Levels and Immunohistochemical Labelling of Des-Gamma-Carboxy-Prothrombin in Patients Undergoing Liver Transplantation for Hepatocellular Carcinoma"

_diagnostics, 2024, doi:10.3390/diagnostics14090894_

Round 1
Reviewer 1 Report (New Reviewer)
Comments and Suggestions for Authors
The authors present an original study entitled “Association of serum levels and immunohistochemical labelling of DCP in patients undergoing liver transplantation for hepatocellular carcinoma”.
The results of the presented study are interesting, novel, and have clinical significance. The article has a good scientific soundness.
However, I have a few important points:
1. In the Results section, "scheme 1" is provided. This scheme is not mentioned in the text and it is most likely appropriate to make it part of Table 1, or Table 1 from the supplement.
2. The issue of statistical analysis of intergroup differences is crucial. The authors use the Kruskal-Wallis test to analyse intergroup differences in quantitative variables between the three groups. However, they do not perform post hoc pairwise comparisons using, for example, Dunn's test. This does not allow us to determine between which groups there were statistically significant differences, as the Kruskal-Wallis test cannot provide an answer to this. Therefore, this statements are not correct – “Serum DCP levels at diagnosis were significantly higher in patients harbouring focally positive (258 mAU/ml) and diffusely positive lesions (257 mAU/ml) compared to candidates with negative labelled lesions (47.5 mAU/ml, table 1); The median maximum diameter was 20 mm (IQR 12-26) in the diffusely positive labelling group compared to 12 mm (IQR 7-21) in the other groups (p=0.002).”
3. Figure 1: 95% confidence interval and p for AUC should be reported.
Author Response
You will find below our point-by-point answers to your comments (in italics).
- In the Results section, "scheme 1" is provided. This scheme is not mentioned in the text and it is most likely appropriate to make it part of Table 1, or Table 1 from the supplement.
Dear Reviewer, we have gone through the Result section multiple times, and, unfortunately, we could not find that a scheme 1 is ever mentioned. Consequently, we are sorry to say that we cannot find a solution for this issue.
- The issue of statistical analysis of intergroup differences is crucial. The authors use the Kruskal-Wallis test to analyse intergroup differences in quantitative variables between the three groups. However, they do not perform post hoc pairwise comparisons using, for example, Dunn's test. This does not allow us to determine between which groups there were statistically significant differences, as the Kruskal-Wallis test cannot provide an answer to this. Therefore, this statements are not correct – “Serum DCP levels at diagnosis were significantly higher in patients harbouring focally positive (258 mAU/ml) and diffusely positive lesions (257 mAU/ml) compared to candidates with negative labelled lesions (47.5 mAU/ml, table 1); The median maximum diameter was 20 mm (IQR 12-26) in the diffusely positive labelling group compared to 12 mm (IQR 7-21) in the other groups (p=0.002).”
Dear Reviewer, we thank you for this suggestion. Indeed, we had not performed Dunn’s tests for the quantitative variables whose distributions were significantly different at Kruskal-Wallis test. Now, we have performed multiple comparisons tests, as is reported in the Methods section. The sentences you mentioned have been amended in the Results section, and even the Discussion has been modified to correctly present our interpretation of these findings.
- Figure 1: 95% confidence interval and p for AUC should be reported.
Dear reviewer, thank you for this suggestion. We have updated the picture and added the 95% confidence intervals and P values for the AUCs.
Reviewer 2 Report (New Reviewer)
Comments and Suggestions for Authors
Authors described as Des-gamma-carboxy-prothrombin (DCP) could be used as an early prognostic marker for hepatocellular cancer (HCC).
The work is very interesting; however I think that is incomplete due to an huge miss.
In my opinion, if authors would like to propose DCP as diagnostic and/or prognostic marker they should demonstrate that it's no detectable, or it's observed with very low levels, in other liver cancers, such as intrahepatic and perihilar cholangiocacinoma. Moreover, authors should demostrante the same things in pre-tumoral conditions such as HBV and HCV infection, chirrosis, NASH and NAFLD. These are some examples but not all.
Based on the idea expressed above, I think that the submitted manuscript is not suitable in the present form.
Author Response
Dear Reviewer, we are truly grateful for your observation. Your proposal will certainly be the matter for a future study, in the wake of researching the role of tissue DCP labelling in primary liver diseases and in primary liver cancers other that hepatocellular carcinoma, which we have acknowledged in the Discussion section. Unfortunately, however, this was not the purpose of our work, where we intended to explore to which extent serum levels and tissue labelling of DCP are linked in hepatocellular carcinoma. Besides, our institution will not consider allocating extra funds on a project that did not originally included primary liver diseases or other primary liver cancers, because the magnitude and length of such a study are drastically increased compared to what we had envisaged and carried out. We understand (and are glad) that a reviewer can suggest pathways for future investigations. At the same time, we are less sure that this should be a point on its own to reject an article. We thank you for your understanding.
Round 2
Reviewer 2 Report (New Reviewer)
Comments and Suggestions for Authors
As indicated in my previuous report, authors have done a good work. However, their claim is too high for the data that they presented. In fact, they affirm: “serum DCP shows good diagnostic accuracy for HCC.” (lines 262-263), but hey have not analyzed any other pre-tumoral conditions or gastrointestinal tumor, e.g. intra-hepatic cholangiocarcinoma.
In my opinion they should reduce their claim and add these limitations or they demonstrate the DCP levels in pre-tumoral and tumoral conditions.
For this reason, I continue to not support the present form of manuscript suitable for the publication.
Author Response
Dear Reviewer, we understand your observation. We have adapted the sentence you mention in order to make it less bombastic.
Round 3
Reviewer 2 Report (New Reviewer)
Comments and Suggestions for Authors
In my opinion the manuscript can be accepted in the present form.
This manuscript is a resubmission of an earlier submission. The following is a list of the peer review reports and author responses from that submission.
Round 1
Reviewer 1 Report
Comments and Suggestions for Authors
The Authors conducted a well-designed prospective study to assess the relationship between preoperative DCP level and liver transplantation outcome in a patient cohort selected within the Milan criteria.
The main limitation of this study is the small number of patients enrolled (39). Nevertheless, the study is well organized, and the manuscript is well written.
I believe the data shown are important because the Authors describe a significant correlation between the DCP preoperative values and the HCC recurrence.
I am unsure if all the demographic data reported in Table 1 is needed. Probably, a number of them might be removed to make the table reading easier.
Author Response
Thank you very for much for your kind remarks.
About table 1 we took out some of the data and created a supplementary table 1 with some data that are still relevant for our study.
Reviewer 2 Report
Comments and Suggestions for Authors
DCP levels are used as a prognostic indicator for HCC. Higher DCP levels are often associated with more aggressive tumor behavior and a poorer prognosis. The work clearly documents, supported by statistics, this correlation and presents arguments for a worse prognosis in patients with elevated DCP levels in the group meeting the Milano criteria. I suggest accepting the paper for publication."
Author Response
Thank you very much for your kind remarks.